# Meta-learning Sparse Implicit Neural Representations

**Jaeho Lee**[A*]     **Jihoon Tack**[B*]     **Namhoon Lee**[CD]     **Jinwoo Shin**[AB]

[A] School of Electrical Engineering, KAIST
[B] Kim Jaechul Graduate School of AI, KAIST
[C] Graduate School of Artificial Intelligence, UNIST
[D] Department of Computer Science and Engineering, UNIST
{jaeho-lee,jihoontack,jinwoos}@kaist.ac.kr, nlee@unist.ac.kr

## Abstract

Implicit neural representations are a promising new avenue of representing general signals by learning a continuous function that, parameterized as a neural network, maps the domain of a signal to its codomain; the mapping from spatial coordinates of an image to its pixel values, for example. Being capable of conveying fine details in a high dimensional signal, unboundedly of its domain, implicit neural representations ensure many advantages over conventional discrete representations. However, the current approach is difficult to scale for a large number of signals or a data set, since learning a neural representation—which is parameter heavy by itself—for each signal individually requires a lot of memory and computations. To address this issue, we propose to leverage a meta-learning approach in combination with network compression under a sparsity constraint, such that it renders a well-initialized sparse parameterization that evolves quickly to represent a set of unseen signals in the subsequent training. We empirically demonstrate that meta-learned sparse neural representations achieve a much smaller loss than dense meta-learned models with the same number of parameters, when trained to fit each signal using the same number of optimization steps.

## 1  Introduction

An explosively growing line of research on implicit neural representations (INRs)—also known as coordinate-based neural representations—studies a new paradigm of signal representation: Instead of storing the signal values corresponding to the coordinate grid (e.g., pixels or voxels), we train a neural network with continuous activation functions (e.g., ReLU, sinusoids) to approximate the coordinate-to-value mapping [31, 26, 3]. As the activation functions are continuous, the trained INRs give a *continuous* representation of the signal. The continuity of INRs brings several practical benefits over the conventional discrete representations, such as providing an out-of-the-box method for superresolution or inpainting tasks [2, 37, 24, 36], or having its number of parameters not strictly scaling with the spatial dimension and/or spatial resolution of the signal [31, 26, 3, 8, 1, 34, 32]. Furthermore, as INRs take a form of trainable models, they are readily amenable to the idea of "learning a prior" from a set of signals, which can be utilized for various purposes including image generation [2, 37, 36] or view synthesis [40, 36, 35, 27].

Despite many advantages, this network-as-a-representation approach is difficult to scale to handle a large set of signals, as having a parameter-heavy neural network trained for each signal requires a lot

---

*Equal contributions
  Code: https://github.com/jaeho-lee/MetaSparseINR

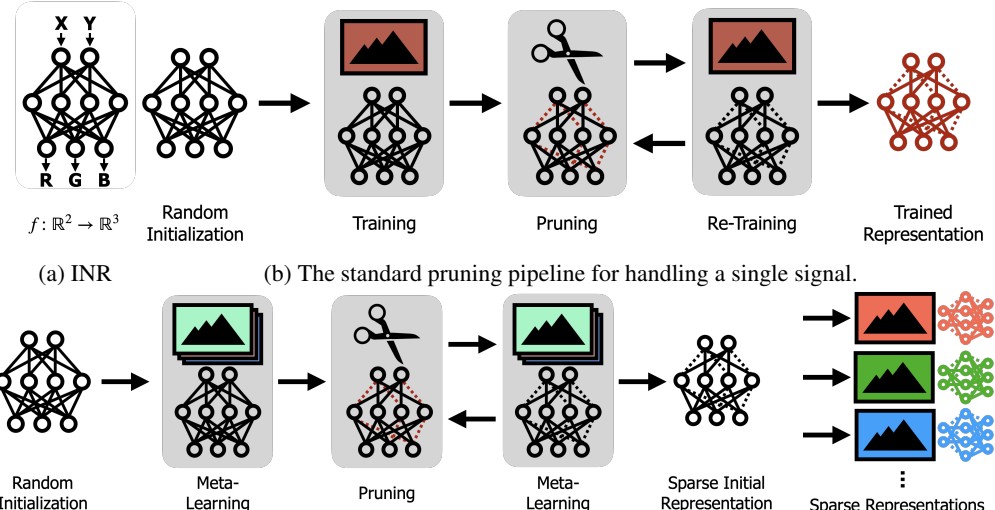

(a) INR      (b) The standard pruning pipeline for handling a single signal.

(c) The proposed Meta-SparseINR procedure for handling multiple signals.

Figure 1: Illustrations of (a) an implicit neural representation, (b) the standard pruning algorithm [9] that prunes and retrains the model for each signal considered, and (c) the proposed Meta-SparseINR procedure to find a sparse initial INR, which can be trained further to fit each signal.

of memory and computations. Existing approaches to address this problem can be roughly divided into three categories. The first category of works utilizes a neural network structure shared across the signals, which takes a latent code vector[2] as an input and modifies the INR output accordingly (e.g., [36, 2, 31]). While these methods only need to store a latent vector for each signal (along with shared model parameters), they have a limited ability to adapt to signals falling outside the learned latent space; this may lead to a degraded representation capability on new signals. The second line of works adopt meta-learning approach to train an *initial INR* from which each signal can be trained to fit within a small number of optimization steps [35, 40]. Unfortunately, these methods do not reduce the memory required to store the neural representations for each signal. The work belonging to the third category [4] reduces the memory required to store the INR model parameters by uniformly quantizing the weights of each INR. The method, however, does not reduce the number of optimization steps needed to train for each signal, or utilize any information from other signals to improve the quantization. In this sense, all three approaches have their own limitations in providing both memory- and compute-efficient method to train INRs for both seen and unseen signals.

To address these limitations, we establish a new framework based on the *neural network pruning*, i.e., removing parameters from the neural representation by fixing them to zero (e.g., [29]). Under this framework, the objective is to train a set of *sparse* INRs that represent a diverse set of signals (either previously seen or unseen) in a compute-efficient manner. To achieve this goal, we make the following contributions:

- We formulate this problem as finding a *sparse initial INR*, which can be trained to fit each signal within a small number of optimization steps. By considering such formulation, we eliminate the need to prune the INRs separately for every single signal, each of which typically requires a computationally heavy prune-retrain cycle (Fig. 1b). We give a detailed formulation in Section 3.
- To solve this newly formulated problem, we propose a pruning algorithm called Meta-SparseINR, which is the first pruning algorithm designed for the INR setups (up to our knowledge). Meta-SparseINR alternately applies (1) a meta-learning step to learn the INR that can be efficiently trained to fit each signal, and (2) a pruning step to removes a fraction of surviving parameters from the INR using the meta-trained weight magnitudes as a saliency score (Fig. 1c). See Section 4 for a complete description of the algorithm.

In Section 5, we validate the performance of the proposed Meta-SparseINR scheme on three natural/synthetic image datasets (CelebA [21], Imagenette [14], SDF [40]) using a widely used INR

---

[2]a low dimensional vector generated by forwarding each signal through some encoder,

architecture (SIREN [36])[3]. In particular, we show that, when trained to fit seen/unseen signals with a small number of optimization steps, Meta-SparseINR consistently achieves a better PSNR than the baseline methods using similar number of parameters, which include the meta-trained dense INRs with narrower width [40] and random pruning. Furthermore, in Section 5.3, we provide results on an exploratory experiment which sheds a light on the potential limitations of *pruning-at-initialization* schemes for fitting multiple signals under the INR context.

## 2 Related works

Instead of giving a general overview on the implicit neural representations and neural network pruning literature, we focus on describing the works that are more directly related to the main ideas of this manuscript. For an overview of two disciplines, we refer the readers to excellent surveys of Tewari et al. [41] and Hoefler et al. [12] on implicit neural representations and the neural network pruning, respectively.

**Fitting implicit neural representations for multiple signals.** As briefly discussed in Section 1, there are at least three different approaches to efficiently train the INRs for a large set of signals. The first approach is to have a neural network component shared across the signals, which uses a latent code (i.e., a low-dimensional vector generated from each signal) to modify the INR output. [36] trains a hypernetwork to predict INR parameters from the latent code, to utilize the knowledge learned from the signal set to fit each signal better. Other works [2, 26, 31] constructs an INR architecture that takes a concatenation of latent code and coordinate vector as an input, to predict the signal values corresponding to the each signal. This approach has advantages in terms of both memory and computation: in terms of parameters, the per-signal memory cost equal to the size of the latent vector, and the per-signal training cost is total training cost of the shared network, divided by the number of samples. These methods, however, have limitations in expressing the signals that has not been observed during the training phase, as pointed out in [40]. The second approach is to meta-learn the initial INR from which each signal can be efficiently fit using a small number of gradient descent steps. [35] proposes an algorithm based on this approach to learn the signed distance function (SDF) corresponding to 2D/3D shapes, and shows that such pre-training by meta-learning significantly reduces the number of optimization steps required to encode new signals. [40] proposes a similar algorithm that works well on a wider range of representation tasks, including 2D image regression. We note that, despite its computational benefits, this approach does not provide any benefit in terms of the parameter-efficiency. The third approach is compress each INR after training. [4] propose one such method based on the weight quantization: They first fit INRs for each signal via extensive training. Then, they quantize the weights from 32-bit precision to 16-bit precision. This method does not provide any advantage in terms of computations, but can potentially be combined with other approaches. The approach taken in the present paper is most related to the second line of works [35, 40]; indeed, one of our main baselines will be the method of [40] used to train dense but narrower INRs.

**Training from sparse initial models.** Recent studies [6, 18, 23] have shown that one can find a sparse subnetwork (i.e., a pruned network) of a randomly initialized model that can closely achieve the performance level of the original dense model when trained from scratch. This finding has brought a surge of interest toward developing algorithms that can find such "winning ticket" subnetworks by masking the original models using various notions of saliency scores (e.g., [44, 38]). Although the research direction has seen a great progress overall, it has been less studied how one can find the best subnetwork when there are multiple datasets at hand. A closely related work is the paper by Morcos et al. [28], which shows that the winning ticket discovered from a dataset can be transferred to a highly relevant dataset. In this paper, the aim is to find a sparse initial model that performs well on *a set of different tasks* (which may be fitting a set of images/3D shapes in this context) when trained further on each task. We note, however, that we consider a slightly different setup than the previous works: We do not restrict ourselves to the models generated by pruning a randomly initialized model, and instead prune a meta-trained model. Indeed, our observation in Section 5.3 suggests that the models generated by pruning a randomly initialized model may not be able to significantly outperform dense and narrow baselines, in the INR context.

---

[3]In Appendix B, we report additional experimental results another INR architecture called Fourier feature networks (FFN [39]), consisting of a Fourier encoding layer and fully-connected ReLU layers thereafter.

**Meta-learning & network pruning.** One of the earliest papers considering the combination of meta-learning with neural network pruning is the work by Liu et al. [22]. In [22], the goal is to train a *PruningNet* which generates a set of channel-pruned architecture and corresponding weights, satisfying the layerwise sparsity requirements given as an input. A more recent paper by Tian et al. [43] is slightly more related to our setting: In [43], the goal is to improve the generalization property of few-shot meta-learning algorithms, without any considerations on the memory or computation limitations; the proposed algorithm alternatively removes and restores the connections from the meta-learned model, as in DSD [10] or SDNN [15]. Unlike these works, the primary aim of this paper is to provide a method to *compress* a set of neural networks; we use meta-learning as a tool to generate a shared model that can be compressed and adapted to each signal. Despite its practical importance, up to our knowledge, we are the first to address this problem.

## 3 A framework for efficiently learning sparse neural representations

The goal of this paper is to develop a memory- and compute-efficient framework for learning and storing the implicit neural representations for a large number signals. We formulate this problem as finding a well-initialized sparse subnetwork structure; the sparse neural representation for each signal can be directly trained from this sparse subnetwork, without requiring a further pruning by computation-heavy prune-retrain cycle (see, e.g., [9]) for each individual signal.

**Formalisms.** For a formal description, we introduce some notations: Let $\mathcal{T} = \{T_1, T_2, \ldots, T_N\}$ be a set of $N$ target signals that we want to approximate. Each signal $T_j : \mathbb{R}^k \to \mathbb{R}^m$ is a function that maps a $k$-dimensional coordinate vector to an $m$-dimensional signal value. For example, a two-dimensional RGB image can be represented by the signal function $T : \mathbb{R}^2 \to \mathbb{R}^3$ with $T(x, y) = (R, G, B)$, and the occupancy of a 3D object (i.e., whether a point in 3D space in occupied by the object) can be represented by the coordinate-to-occupancy function $T : \mathbb{R}^3 \to \{0, 1\} \subset \mathbb{R}$. We assume that we have an access to data pairs $\{\mathbf{x}_i, T_j(\mathbf{x}_i)\}_{i \in \mathcal{I}}$ for each signal $j \in \{1, \ldots, N\}$, where $\{\mathbf{x}_i\}_{i \in \mathcal{I}}$ denotes the finite set of predefined input coordinate vectors indexed by $\mathcal{I}$. For two-dimensional $256 \times 256$ images, for instance, the input coordinate vectors may be 2D vectors belonging to the grid $\{0, 1/255, 2/255, \ldots, 254/255, 1\}^2$.

We use the implicit neural representation $f(\cdot; \theta) : \mathbb{R}^k \to \mathbb{R}^m$ with a trainable parameter vector $\theta \in \mathbb{R}^d$ to approximate each signal. For SIRENs [36], $f(\cdot; \theta)$ will be a multi-layer perceptron (MLP) with sinusoidal activation functions; for Fourier feature networks (FFN [39]), $f(\cdot; \theta)$ will be an MLP with random Fourier feature encoders and ReLU activation functions. Now, the *representation risk* of approximating $j$-th signal with a parameter $\theta$ can be written as

$$\mathcal{L}_j(\theta) = \sum_{i \in \mathcal{I}} \|f(\mathbf{x}_i; \theta) - T_j(\mathbf{x}_i)\|_2^2. \tag{1}$$

Note that, while we are using the squared loss for a simple presentation, any other loss can be used (e.g., $\ell_1$ loss). Under typical scenarios without any sparsity considerations, one may optimize the parameters independently for each signal. In other words, we solve the optimization problem

$$\min_{\theta_1, \ldots, \theta_N \in \mathbb{R}^d} \frac{1}{N} \sum_{j=1}^N \mathcal{L}_j(\theta_j) = \frac{1}{N} \sum_{j=1}^N \left( \min_{\theta_j \in \mathbb{R}^d} \mathcal{L}_j(\theta_j) \right), \tag{2}$$

which can be decomposed into $N$ subproblems. In this case, the total number of parameters is $dN$.

**Sparse subnetwork and initialization.** To reduce the number of parameters, we consider finding a shared sparse subnetwork structure having at most $\kappa < d$ connections, so that the total number of parameters is reduced to $\kappa N$. More formally, the subnetwork structure can be expressed as a binary *mask* $M \in \{0, 1\}^d$ with $\|M\|_0 \leq \kappa$, which is applied to the neural representation by taking an element-wise product with the parameter, i.e., $f(\mathbf{x}; M \odot \theta)$. We hope to find a zero-one mask vector $M$ that solves the optimization

$$\min_{\substack{M \in \{0,1\}^d \\ \|M\|_0 \leq \kappa}} \min_{\theta_1, \ldots, \theta_N \in \mathbb{R}^d} \frac{1}{N} \sum_{j=1}^N \mathcal{L}_j(M \odot \theta_j) = \min_{\substack{M \in \{0,1\}^d \\ \|M\|_0 \leq \kappa}} \frac{1}{N} \sum_{j=1}^N \left( \min_{\theta_j \in \mathbb{R}^d} \mathcal{L}_j(M \odot \theta_j) \right). \tag{3}$$

The problem (3), however, needs to be refined for practical considerations. To solve each subproblem in Eq. (3) without too many gradient computations, we also require a good initialization. Indeed,

it has been known that without a well-chosen initialization, the optimization of sparse networks often takes significantly longer [20], or even fail to achieve the performance level that can be achieved with a well-chosen one [6, 19].

To this end, we augment the problem (3) with the notion of *optimized initialization* as follows. Let $\theta_j^{(t)}(M; \theta^{(0)}) \in \mathbb{R}^d$ denote the parameter that has been attained by taking $t$ steps of gradient-based optimization (e.g., SGD or Adam) to fit the target signal $T_j$, on the subnetwork with an initial value $\theta^{(0)}$ and masked by $M$. Given some steps-to- budget $t$, we want to find a pair of mask $M \in \{0, 1\}^d$ and the initial value $\theta^{(0)} \in \mathbb{R}^d$ that jointly solves

$$\min_{\theta^{(0)} \in \mathbb{R}^d} \min_{\substack{M \in \{0,1\}^d \\ \|M\|_0 \le \kappa}} \frac{1}{N} \sum_{j=1}^{N} \mathcal{L}_j \left( M \odot \theta^{(t)}(T_j, \theta^{(0)}) \right). \tag{4}$$

We propose a meta-learning approach to solve the problem (4) in Section 4.

Before we move on, we note that the problem cannot be directly solved—without an architectural change—by simply *pooling* the data pairs from $N$ signals and applying standard pruning techniques (e.g., [9]). This is because we are considering the task of signal representation, instead of a classification (or similar). Under the signal representation setup, the data pairs from different signals share the coordinate input space $\{\mathbf{x}_i\}_{i \in \mathcal{I}}$. If data pairs from different signals are pooled, this leads to conflicting pairs of samples that have different output values for the same input.

## 4 Learning sparse neural representations with meta-learning

As described in Eq. (4), our goal is to find a well-initialized sparse implicit neural representation from which the neural representations for each signal can be efficiently trained. To achieve this goal, we take a meta-learning approach: We meta-learn an implicit neural representation that can quickly adapt to a set of signals, and then prune the model based on its weight magnitudes (see Algorithm 1). We call this procedure Meta-SparseINR (Meta-learning Sparse Implicit Neural Representation). For completeness, we provide preliminaries on the meta-learning procedure we use in Section 4.1, and give a full description of the Meta-SparseINR algorithm in Section 4.2.

### 4.1 Preliminaries: Meta-learning with neural networks

Meta-learning [42] aims to give models that can be trained efficiently on a given task, by utilizing a form of knowledge gained from training on a family of relevant but different tasks. This knowledge can be transferred in various ways, e.g., utilizing neural network structures with memory-augmentations [33]. In this section, however, we focus on describing the *model-agnostic* approach that explicitly trains model parameters to convey the learned knowledge (namely, MAML [5]), which we use as a part of our method. This is because (i) it allows us to use popular implicit neural representation architectures (e.g., [36, 39]) without modifications, and (ii) we will use the model parameters as a saliency score for pruning, as will be described in the following subsection.

While MAML is not originally proposed for the signal representation, we describe MAML using the INR setup and notations (introduced in Section 3) for simplicity: Given a set of signals $\mathcal{T} = \{T_1, \ldots, T_N\}$, MAML [5] aims to learn model parameters that can adapt to each signal within a small number of gradient steps. To do so, one performs the following iterative optimization: At each *outer loop*, the learner draws one (or more) signal from the signal set. For now, suppose that this is the $j$-th iteration of the outer loop and only one signal has been drawn, which we denote by $T_j$. Then, the learner computes a $t$-step SGD-updated version of the parameter $\theta^{(t)}(T_j, \theta_j)$ from the current parameter $\theta_j$. Then, one updates the parameter $\theta_j$ by taking a stochastic gradient descent (SGD) with some outer step size $\beta$:

$$\theta_{j+1} \leftarrow \theta_j - \beta \cdot \nabla_{\theta_j} \mathcal{L}_j \left( \theta^{(t)}(T_j, \theta_j) \right), \tag{5}$$

where $\mathcal{L}_j$ denotes the representation risk computed for the signal $T_j$, as in Section 3. This ends a single outer step. We denote running $\tau$ steps of outer loops on the current parameter $\theta$, using the signal set $\mathcal{T}$, outer step size $\beta$, number of inner SGD steps $t$ by $\texttt{MAML}_{t,\beta}^{(\tau)}(\theta; \mathcal{T})$. In our method, we will also perform MAML on the sparse networks (masked by some $M$). In this case, we will denote the same procedure by $\texttt{MAML}_{t,\beta}^{(\tau)}(\theta; M, \mathcal{T})$.

---

**Algorithm 1** Meta-SparseINR: Meta-learning Sparse Implicit Neural Representations

---

**Require:** Signal set $\mathcal{T} = \{T_1, \ldots, T_N\}$, desired sparsity $\kappa$, per-iteration pruning ratio $\gamma$,
        number of pre-/re-training steps $\tau, \tilde{\tau}$, number of inner SGD steps $t$,
        outer loop step size $\beta$.

1: Initialize $\theta \in \mathbb{R}^d$ using the standard initialization scheme.
2: Initialize mask $M = (1, 1, \ldots, 1)$ of length $d$.            ▷ Begin from the unpruned network.
3: $\theta \leftarrow \texttt{MAML}_{t,\beta}^{(\tau)}(\theta; \mathcal{T})$.            ▷ Pretraining. See Section 4.1 for details.
4: **while** $\|M\|_0 > \kappa$ **do**
5:      $\mathbf{s}_i = M_i \cdot |\theta_i|, \quad \forall i \in \{1, \ldots, d\}$.            ▷ Compute the magnitude score.
6:      $\tilde{\mathbf{s}} = \text{SortDescending}(\mathbf{s})$
7:      $M_i \leftarrow \mathbf{1}\{\mathbf{s}_i \geq \tilde{\mathbf{s}}_{\gamma \cdot \|M\|_0}\}, \quad \forall i \in \{1, \ldots, d\}$            ▷ Prune connections by the magnitude.
8:      $\theta \leftarrow \texttt{MAML}_{t,\beta}^{(\tilde{\tau})}(\theta; M, \mathcal{T})$.            ▷ Retraining.
9: **end while**

---

### 4.2 Meta-SparseINR: Meta-learning sparse implicit neural representations

We now describe our method, Meta-SparseINR (Algorithm 1), to learn the sparse implicit neural representation that can be efficiently trained to fit each signals. In a nutshell, Meta-SparseINR operates by alternately meta-training the INR using MAML, and pruning the INR using the magnitude-based pruning; this procedure can be viewed as an alternating minimization of Eq. (4), where meta-training steps and pruning steps approximately perform the outer and inner minimization steps in Eq. (4), respectively. More concretely, Meta-SparseINR consists of three steps:

**Step 1 (Meta-learning).** Given a set of signals and an INR randomly initialized with standard initialization schemes (e.g., as in [36] for SIRENs), we meta-learn the INR over the signals by running MAML for $\tau$ steps.

**Step 2 (Pruning).** From the learned INR, we remove $\kappa\%$ of surviving connections with smallest weight magnitudes. We use the *global* magnitude criterion, i.e., we apply a single magnitude threshold for every layers instead of having thresholds for each layer.

**Step 3 (Retrain and repeat).** We retrain the pruned INR by running MAML for $\tilde{\tau}$ steps; $\tilde{\tau}$ required to converge to the desired performance level may often be smaller than $\tau$ used in Step 1. If the desired global sparsity level has not been met, go back to Step 2.

As we will validate through our experiments in Section 5, meta-learning is an essential component in Meta-SparseINR. More specifically, meta-learning steps play, at least, a twofold role: First, the weights given by the meta-learning can be trained to fit each signal with a much smaller number of optimization steps. This fact may not come as a big surprise, as the meta-learning objective (5) explicitly optimizes the *weight after optimizing for a few steps*. Indeed, without any pruning considerations, this point has been validated by Tancik et al. [40]. We observe that the same holds for sparse INRs, as we will observe in our main experiments in Section 5.2.

Second, the magnitude of the weights learned by meta-learning acts as a powerful *saliency score* for the pruning step (Step 2). Intuitively, the connections with a large meta-learned weight magnitudes are likely to have a relatively large weight magnitude after a $t$-step adaptation to a particular signal (with a small $t$). In this sense, the meta-learned weight magnitudes can be viewed as an *aggregation* of the weight magnitudes for each signal. As it is believed that weight magnitudes are an effective saliency criterion for the model trained for a single task (or signal, in this context), we expect the magnitudes of the meta-learned weights to be an effective saliency criterion for the signals, on average.[4] Indeed, we show in Section 5.2 that removing the connections with the smallest magnitude of the meta-learned weights outperforms both random pruning and dense, narrow INRs.

---

[4] Aside from this intuition, the magnitude criterion is much more computationally efficient than its competitors, such as Hessian-based ones [17, 11]; Hessian-based ones may require the evaluation of a third-order derivative, as we are using the meta-learning loss.

In addition to the second point, we note that we make an empirical observation which suggests that starting from the (meta-)trained weights may even be necessary for sparse models to outperform the dense, narrow models in the INR context. We give more details in Section 5.3.

## 5 Experiments

In this section, we experimentally validate the performance of the proposed Meta-SparseINR (see Algorithm 1) by measuring its performance on various 2D regression tasks. We describe the experimental setup and the baseline methods Section 5.1. We report the experimental results and discuss the observations in Section 5.2. In Section 5.3, we describe the experimental setup and results for an additional exploratory experiment; the experiment is designed to gauge the potential of "pruning-at-initialization" schemes to learn sparse INR that can be efficiently trained to fit each signal.

### 5.1 Experimental setup

We evaluate the performance of the proposed Meta-SparseINR under the following two-phase scenario: We are given a set of signals with training and validation splits. During the training phase, we meta-learn the initial INR from the training split of signals. During the test phase, we randomly draw 100 signals from the validation split of signals, and train for 100 steps using the full-batch Adam to fit the INR on each signal.[5] We report the average performance (PSNR) of the model over these 100 samples. We average over three seeds for all experiments in the following subsection.

**Datasets.** We focus on 2D image regression tasks using three image datasets with distinct characteristics: CelebA (face data [21]), Imagenette (natural image [14]), and 2D SDF (geometric pattern [40]). We use the default training and validation splits. All images are resized and cropped to have the resolution of $178 \times 178$.

**Models.** As our base INR, we use multi-layer perceptrons (MLPs) with sinusoidal activation functions (SIREN [36]), having four hidden layers with 256 neurons in each layer. This configuration is popularly used for the 2D image regression task in previous works, e.g., [36, 40]. In Appendix B, we report additional experimental results on the another widely used class of architectures: MLPs using random Fourier features (FFN [39]).

**Method & baselines.** We evaluate the proposed Meta-SparseINR along with five baselines.

- *Meta-SparseINR (ours)*: The method proposed in this paper (see Algorithm 1). Throughout the experiments, we prune 20% of the weight parameters in every pruning phase (i.e., $\gamma = 0.2$).
- *Random Pruning*: Same as Meta-SparseINR, but we randomly prune the INR parameters with uniform probability, instead of using the magnitude criterion.
- *Dense-Narrow*: We meta-learn the dense neural representation that has a narrower width than the original INR, as in [40]. The widths are selected to ensure that each sparse INR has a dense-narrow counterpart with an equal or slightly larger number of total parameters.
- *MAML+OneShot*: We fit a MAML-trained model on each signal for 50 epochs. Then, we perform magnitude-based pruning at a single shot, and run additional 50 epochs of training.
- *MAML+IMP*: We prune a MAML-trained model $t$ times, removing 20% of the surviving weights each time, and running $\frac{100}{t+1}$ epochs before/between/after each pruning phase.
- *Scratch*: Same as Dense-Narrow, but we do not meta-learn the weights and use the weights initialized using the standard random initialization scheme (e.g., the one in [36] for SIRENs).

**Other details.** For other experimental details, including the data pre-processing steps and hyperparameters for the meta-learning phases, see Appendix A.

### 5.2 Main experiments

In Fig. 2, we observe that the proposed Meta-SparseINR achieves a consistent performance gain over the all baseline methods for all three datasets. We observe that the gain is larger for in the high-sparsity regime, i.e., for INRs having a much smaller number of parameters. In particular, for

---

[5]We ensure that we use same 100 signals for our method and the baselines.

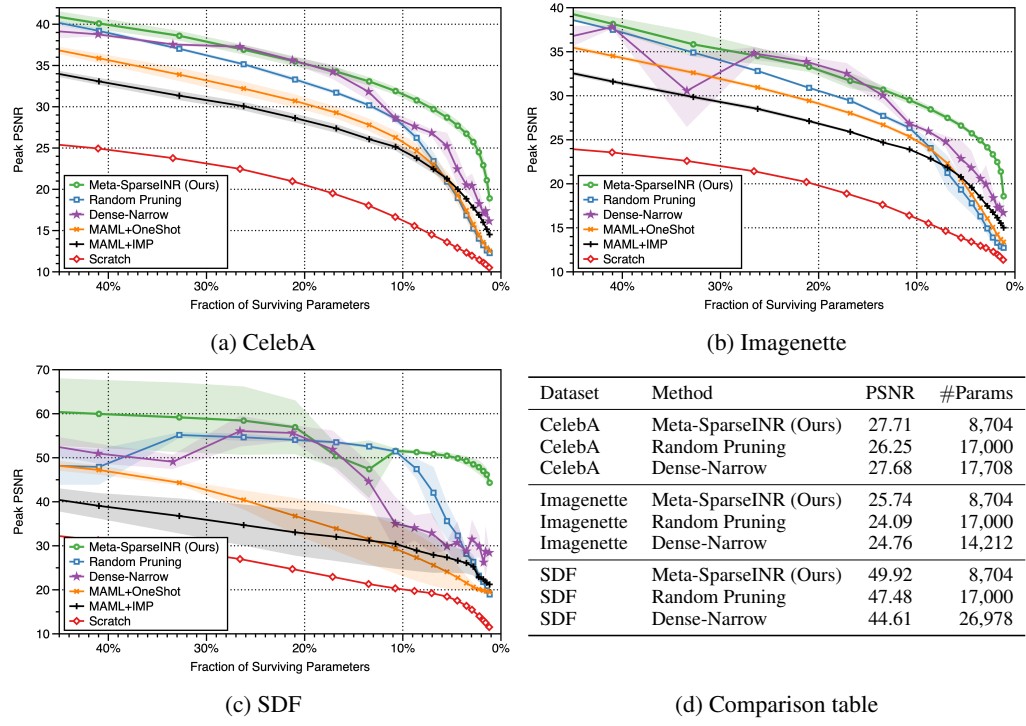

(a) CelebA

(b) Imagenette

(c) SDF

(d) Comparison table

| Dataset | Method | PSNR | #Params |
|---------|--------|------|---------|
| CelebA | Meta-SparseINR (Ours) | 27.71 | 8,704 |
| CelebA | Random Pruning | 26.25 | 17,000 |
| CelebA | Dense-Narrow | 27.68 | 17,708 |
| Imagenette | Meta-SparseINR (Ours) | 25.74 | 8,704 |
| Imagenette | Random Pruning | 24.09 | 17,000 |
| Imagenette | Dense-Narrow | 24.76 | 14,212 |
| SDF | Meta-SparseINR (Ours) | 49.92 | 8,704 |
| SDF | Random Pruning | 47.48 | 17,000 |
| SDF | Dense-Narrow | 44.61 | 26,978 |

Figure 2: Comparison of the average PSNR of meta-learned SIRENs after 100 steps of per-signal training. (a–c) PSNR plots for CelebA, Imagenette, and SDF. (d) Comparing the Meta-SparseINR having 8,704 parameters with strongest baselines (Random Pruning and Dense-Narrow) achieving a similar level of PSNR. Shaded region: Mean $\pm$ one standard deviation (three seeds).

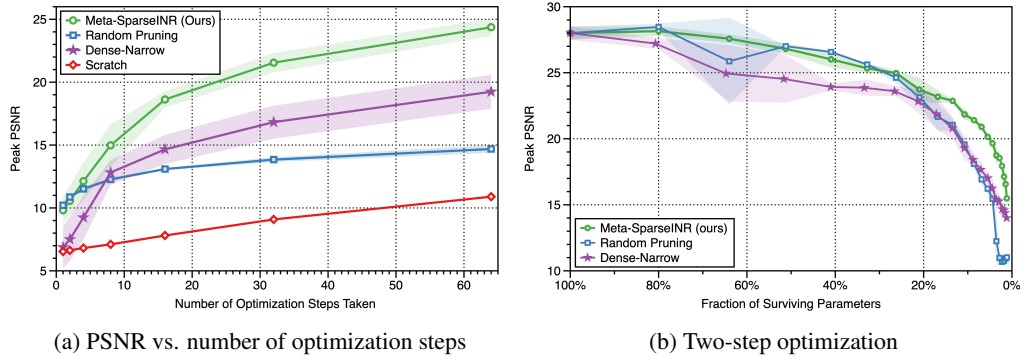

(a) PSNR vs. number of optimization steps

(b) Two-step optimization

Figure 3: Fitting SIRENs meta-learned on CelebA to validation data with a smaller training budget. (a) PSNR trajectory for sparse models having 6,964 parameters and dense-narrow models having 7,152 parameters. (b) PSNR after two steps of training on each sample.

CelebA dataset on SIREN (Fig. 2a), Meta-SparseINR achieves the PSNR $\approx 26.73$ with only 6,964 parameters, whereas the strongest baseline (Dense-Narrow) achieves $\approx 20.52$ with slightly more 7,152 parameters; this is $\times 4.18$ difference in terms of the MSE loss. Among baselines, we observe that Random Pruning and Dense-Narrow are stronger than MAML+Oneshot and MAML+IMP. This suggests that under the per-signal training budget constraints, it is beneficial to have a compressed initial model to start from, instead of compressing the model while fitting each signal.

In Fig. 3, we compare the performance of the proposed Meta-SparseINR with the strongest baselines (Random Pruning and Dense-Narrow) with a per-signal training budget smaller than 100 steps. We observe that Meta-SparseINR still outperforms the baselines in such cases. Interestingly, Dense-Narrow performs worse than Random Pruning at the early stage of per-signal fitting, but achieves

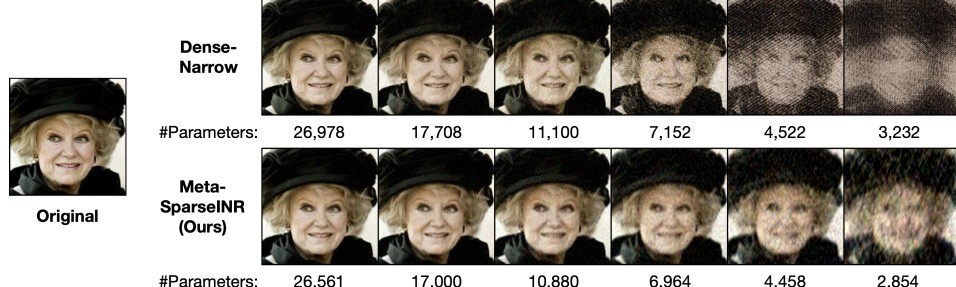

| | Dense-Narrow | | | | | |
|---|---|---|---|---|---|---|
| #Parameters: | 26,978 | 17,708 | 11,100 | 7,152 | 4,522 | 3,232 |
| Meta-SparseINR (Ours) | | | | | | |
| #Parameters: | 26,561 | 17,000 | 10,880 | 6,964 | 4,458 | 2,854 |

Figure 4: A visual comparison of meta-learned SIRENs after training for 100 steps to fit an image. **(Upper row)** Trained from a meta-learned dense and narrow INR, as in [40]. **(Lower row)** Trained from a sparse model learned by the Meta-SparseINR procedure. Sparse models retain more color information and give coarse patterns, while dense-narrow models are more monochromatic with less-structured patterns. We observed a similar behaviors in other examples (see Appendix C).

a better performance than random sparsity as the per-signal training budget increases. From this observation, we suspect that the drawbacks of dense models comes from their expressive power, rather than how efficiently they can be trained.

To better understand the effect of sparsity (as opposed to dense), we compare the distorted images generated by dense and narrow models with images generated by sparse models (Fig. 4). Here, we use the first image from the CelebA validation dataset. We observe that dense-narrow models tend to be more monochromatic and have finer patterns than wide-sparse models. Indeed, we observe a similar trend in other figures as well; see Appendix C for more images.

Lastly, we note that Meta-SparseINR outperforms the baselines in terms of the PSNR achieved for *training split* data as well. Indeed, the PSNR achieved for the training and validation splits differ only by a tiny margin. We report the results on the training split in Appendix D.

## 5.3 Pruning without trained initial weights

In this subsection, we ask whether it is possible to generate a sparse initial INR that can fit a set of signals efficiently, by simply pruning a randomly initialized INR directly for each signal without any trained weights. For this purpose, we compare the performance of dense and narrow models with sparse initial INRs generated by the *winning ticket* method [6], which is known to achieve the best overall performance among pruning-at-initialization schemes [7].

**Setup.** We consider fitting a set of $512 \times 512$ natural images with RGB color channels. For the winning ticket method, we use *iterative magnitude pruning* with weight rewinding [6], with a global magnitude criterion. As in main experiments, we remove 20% of the surviving weights in every iteration. We train for 50k steps with full batch, using Adam with learning rate $1.0 \times 10^{-4}$. For other experimental details (e.g., the image used and hyperparameters), see Appendix A.1.

**Result.** In Fig. 5, we report the PSNR achieved by both methods on SIREN (Fig. 5a) and FFN (Fig. 5b); we report the peak PSNR instead of the final PSNR, as PSNRs tend to be unstable during the end of the training.

We make two observations. First, we observe that the performance of both schemes drop consistently as we consider a smaller number of parameters. Second, none of the two methods noticeably outperform the other. Together with the results in Section 5.2, the observations highlight the importance of the training step preceding the pruning step; meta-learning gives rise to a sparse structure that can be trained to a higher PSNR than the dense-narrow counterpart. This sheds a pessimistic light on the existence of an algorithm that can prune a randomly initialized INR to give a sparse INR that can be trained to fit mutliple signals better than a dense-narrow INR, as the best known pruning-at-initialization scheme does not provide any gain even for a single signal case. As a minor remark, we note that pruning at initialization may have some benefits over dense-narrow models at the early stage of training, while at such stages the PSNR level is generally low, and the benefits vanish as PSNR starts to exceed 20 (at around epoch 1000).

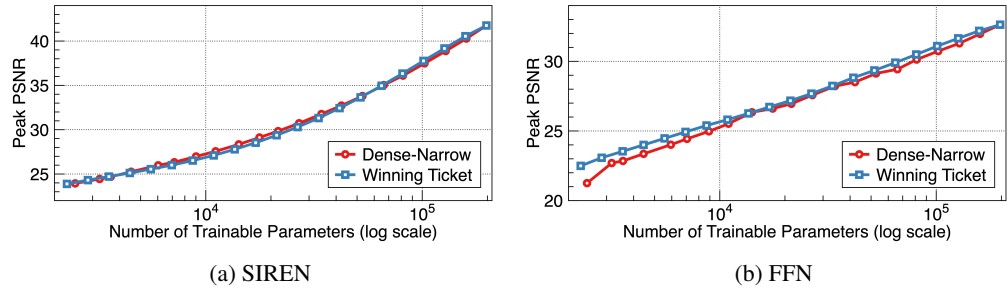

| (a) SIREN | (b) FFN |

Figure 5: Size-PSNR tradeoff curves for fitting a $512 \times 512$ image using dense and narrow INRs (red circles) and wide-sparse INRs pruned at initialization using the "winning ticket" method (blue squares). In both plots, wide-sparse INRs do not perform noticeably well against dense and narrow ones, highlighting the importance of trained weights to enjoy the benefit of sparsity.

## 6    Discussion & limitations

In this paper, we proposed a framework for an efficient training of implicit neural representations, in terms of memory and computation. We formulated the problem as learning a sparse INR that can be trained further to fit each signal within a small training budget, and proposed a first algorithm to solve the problem, called Meta-SparseINR; the proposed algorithm consistently outperforms the baselines in all experimental setups considered.

Adopting the neural network pruning approach, we have implicitly limited the discussion to methods that (1) generate sparse parameters, and (2) does not have any *shared parameters* available. While such properties can come advantageous for some applications (e.g., smaller inference cost), such restrictions are not necessary for improving the parameter efficiency. For instance, one may consider fitting the signals with INRs having a *sparse differences* from a dense, shared reference model; the memory cost to store the shared model will become negligibly small as the number of signals increases. Exploring the potential of such algorithms is a promising future direction.

A potential ethical side-effect of network compression algorithms is their disparate impact on various subgroups [13]. While such impact takes a form of disparate accuracy drop on subgroups, the impact can take a more nuanced and hard-to-detect form in the context of INR training. More specifically, we can think of two additional dimensions of disparate impact: the discrepancy in the training time required to arrive at a desired performance level, and *structurally-biased distortion* in the represented image, e.g., as in [25]. Such potential negative impacts on fairness should be carefully measured, disclosed, and mitigated before being considered for a public use.

## Acknowledgments and Disclosure of Funding

We thank Matthew Tancik for initial suggestions on the model and hyperparameter selection, and thank Kwonyoung Ryu and Sihyun Yu for various technical suggestions on the PyTorch implementation of implicit neural representations.

This work was partially supported by Institute of Information & Communications Technology Planning & Evaluation (IITP) grant funded by the Korea government (MSIT) (No.2019-0-00075, Artificial Intelligence Graduate School Program (KAIST), and No.2020-0-01336, Artificial Intelligence Graduate School Program (UNIST)). This work was mainly supported by Samsung Research Funding & Incubation Center of Samsung Electronics under Project Number SRFCIT1902-06.

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
