# A   Detailed experimental setups

Here, we give additional experimental details not noted in the main text.

**Model.** We use SIREN with the frequency factor $\omega_0 = 200.0$, as in Tancik et al. [40].

**Dataset.** All image datasets used in the main experiments are resized to the resolution of $178 \times 178$. For CelebA, we center-crop to fit this size. For Imagenette, we resize each image to $200 \times 200$, and then cneter-crop. SDF dataset is already of this size. We use the default training and validation splits for each dataset.

**Optimization.** During the initial meta-learning phase of Meta-SparseINR (before pruning), we use the following hyperparameters: For the outer-loop optimization, we use Adam with learning rate $\beta = 1.0 \times 10^{-5}$, and train for $\tau = 150,000$ outer steps. We sample three images for each outer loop, and take $t = 2$ inner SGD optimization steps with the learning rate $1.0 \times 10^{-3}$. For the post-pruning meta-learning phase, we use the same optimization options, but train for $\tilde{\tau} = 30,000$ outer steps. For fine-tuning on each image, we use Adam to train for 100 steps with learning rate $1.0 \times 10^{-3}$. In all experiments, we use full batch.

**Narrower widths.** For Dense-Narrow and Scratch baselines, we experiment on the widths

$$\{256, 230, 206, 184, 164, 148, 132, 118, 106, 94, 84, 76, 68, 60, 54, 48, 44, 38, 34, 32, 28\}. \tag{6}$$

We selected these widths based on two principles: (1) For each sparse INR, there should exist a narrow INR with the same or slightly more number of parameters. (2) Each width should be an even number, so that we can use the same width for MLPs with random Fourier features [39].

**Computational resource.** All experiments have taken place on a server equipped with 40 CPUs of Intel Xeon E5-2630v4  2.20GHz and 8 GPUs of GeForce RTX 2080.

**Other details.** We refer the reviewers for the code attached as the supplementary material.

## A.1   Pruning without trained initial weights

Here, we give addiitonal experimental details for the experiment described in Section 5.3. We refer to Frankle and Carbin [6] for the description of the winning ticket algorithm.

**Winning ticket algorithm.** The winning ticket algorithm consists of four steps: (1) Initialize the model using the standard initialization scheme. (2) Train the model. (3) Prune $\kappa\%$ of the surviving parameters from the model, using the magnitude criterion. (4) Restore the initial model, and go back to step 2. For each training step, we use Adam with learning rate $1.0 \times 10^{-4}$ for 50,000 steps; we use the same schedule for training dense-narrow INRs. We refer the readers to Frankle and Carbin [6] for a more complete description of the winning ticket algorithm.

**Image.** We use $512 \times 512$ center-cropped versions of natural images from the Kodak dataset [16]; we use the first 24 images excluding the last sample, as the height of the last image is smaller than 512 pixels.

**Model.** For SIREN, we use $\omega_0 = 30.0$ as in [36]. For FFN, we use $\sigma = 20.0$.

# B  Experimental results on MLPs using random Fourier features

In this section, we report additional experimental results using the MLPs using random Fourier features (FFN [39]). In particular, we construct an FFN with four hidden layers of width 256, where the first hidden layer uses the sinusoidal activation functions as described in Tancik et al. [39]. For this first layer, we use fixed weight parameters independently drawn from the Gaussian distribution with some standard deviation $\sigma = 20$; under our setup, $\sigma = 20$ performed better than $\sigma = 10$ [39] or $\sigma = 30$ [40]. We note that, as these first layer parameters are fixed (and thus shared over the models), we only prune the parameters in the succeeding layers. All other setups are identical to the SIREN experiment. The experimental results are given in Table 1. As in the SIREN experiments, we observe that Meta-SparseINR outperforms the baselines, while FFN models achieve a slighly lower level of PSNR than SIREN models.

Table 1: Average PSNR of meta-learned FFNs after 100 steps of per-signal training, on CelebA dataset (averaged over three seeds). Fully dense meta-learned FFN achieved $25.41_{\pm1.95}$, and the fully dense randomly initialized FFN achieved $16.85_{\pm0.14}$.

| Surviving Parameters | 80.0% | 64.0% | 51.2% | 41.0% | 32.8% | 26.2% | 21.0% | 16.8% | 13.4% | 10.74% |
|---|---|---|---|---|---|---|---|---|---|---|
| Meta-SparseINR (Ours) | $24.88_{\pm1.62}$ | $24.70_{\pm1.11}$ | $24.69_{\pm0.14}$ | $23.91_{\pm0.23}$ | $23.03_{\pm0.32}$ | $22.12_{\pm0.19}$ | $21.19_{\pm0.22}$ | $19.13_{\pm1.06}$ | $18.09_{\pm1.83}$ | $18.41_{\pm0.38}$ |
| Random Pruning | $25.15_{\pm0.20}$ | $24.05_{\pm0.25}$ | $22.80_{\pm0.23}$ | $21.54_{\pm0.17}$ | $20.02_{\pm0.27}$ | $18.88_{\pm0.26}$ | $18.06_{\pm0.36}$ | $17.20_{\pm0.26}$ | $16.51_{\pm0.36}$ | $15.88_{\pm0.23}$ |

| Surviving Parameters | 80.8% | 64.8% | 51.7% | 41.1% | 33.5% | 26.7% | 25.1% | 20.6% | 16.6% | 13.00% |
|---|---|---|---|---|---|---|---|---|---|---|
| Dense-Narrow. | $22.27_{\pm2.55}$ | $21.36_{\pm2.06}$ | $20.80_{\pm2.61}$ | $19.39_{\pm2.37}$ | $18.04_{\pm2.41}$ | $17.87_{\pm2.87}$ | $16.62_{\pm2.35}$ | $15.90_{\pm2.21}$ | $15.11_{\pm2.12}$ | $14.95_{\pm1.76}$ |
| Scratch. | $15.82_{\pm0.04}$ | $15.02_{\pm0.00}$ | $14.23_{\pm0.03}$ | $13.64_{\pm0.05}$ | $13.09_{\pm0.03}$ | $12.59_{\pm0.03}$ | $12.22_{\pm0.05}$ | $11.84_{\pm0.03}$ | $11.56_{\pm0.03}$ | $11.22_{\pm0.03}$ |

# C   Additional example figures

In Section 5.2, we report a figure (Fig. 4) illustrating the difference between the compressed images that can be learned from dense but narrow meta-learned INRs and Meta-SparseINRs. In particular, we have observed that Dense-Narrow INRs tend to learn more monochromatic images with fine textures, and Meta-SparseINRs tend to learn more coarse-textured images having more color informations. In Fig. 6, we provide additional examples that solidify this observation. We observe a similar behavior on additional samples on CelebA, and also other datasets.

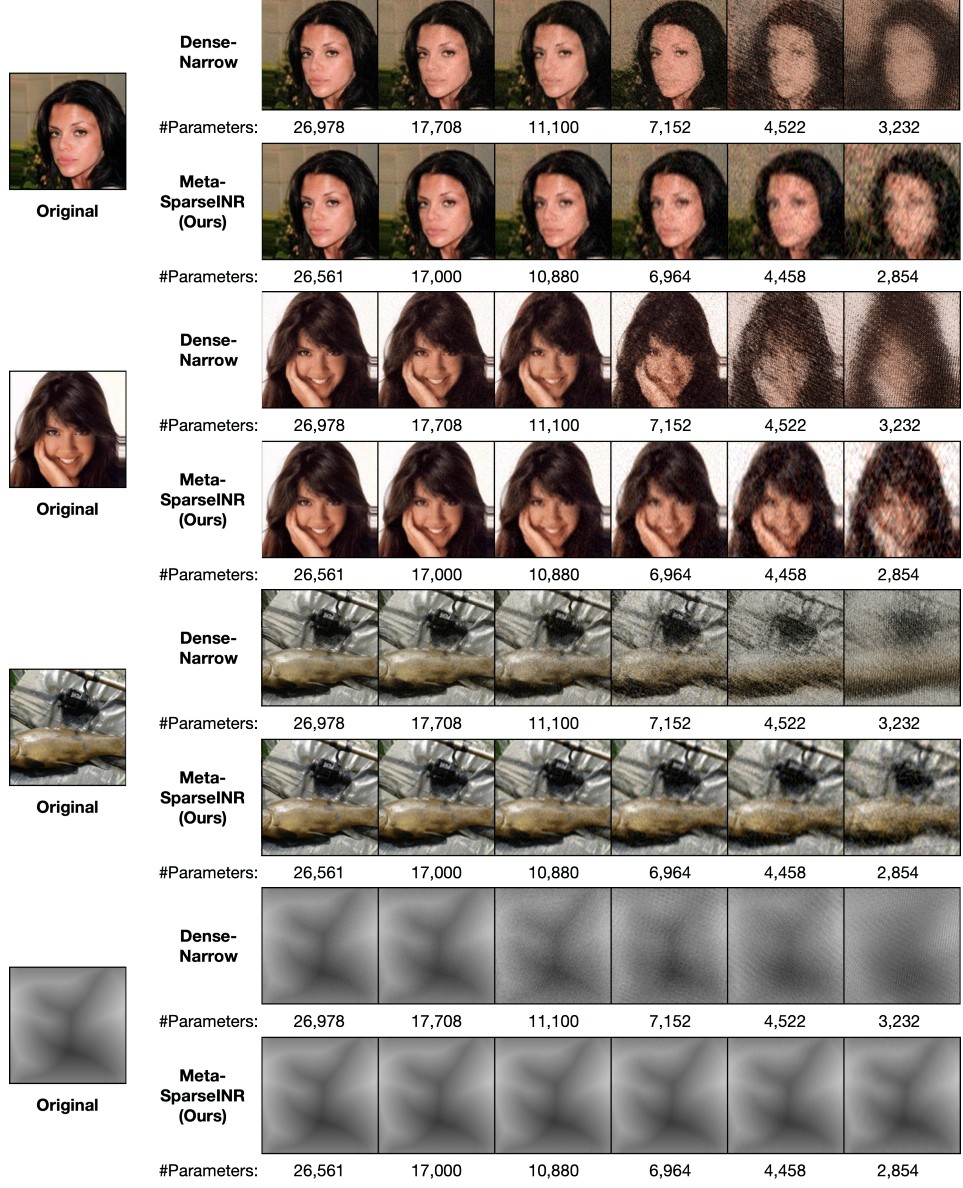

Figure 6: Additional visual comparison of meta-learned SIRENs after training for 100 steps to fit an image. Upper two images are second and third images from the validation split of the CelebA dataset (first is given in Fig. 4), and the lower two images are the first images from Imagenette and SDF dataset, respectively.

# D   Experimental results on the training split

In Section 5.2, we report PSNRs of meta-learned SIRENs after 100-step training on each sample from the validation split of the considered dataset. In Fig. 7, we additionally report PSNRs of fitting each sample in the training split, which has been used to train the meta-learned models. We report this to demonstrate that the proposed problem framework of training a compressed initial INR (as described in Section 3) enables an efficient generalization to unseen samples. Indeed, we observe that there is no notable difference between the PSNRs achieved on the training split and the validation split.

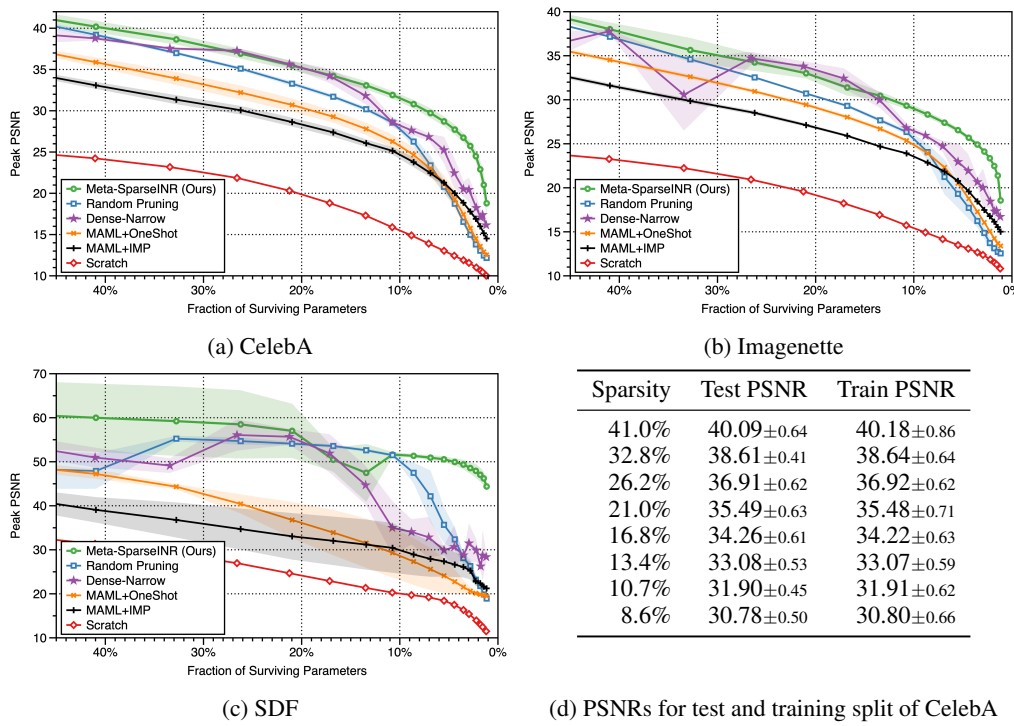

(a) CelebA

(b) Imagenette

(c) SDF

(d) PSNRs for test and training split of CelebA

| Sparsity | Test PSNR | Train PSNR |
|---|---|---|
| 41.0% | $40.09_{\pm 0.64}$ | $40.18_{\pm 0.86}$ |
| 32.8% | $38.61_{\pm 0.41}$ | $38.64_{\pm 0.64}$ |
| 26.2% | $36.91_{\pm 0.62}$ | $36.92_{\pm 0.62}$ |
| 21.0% | $35.49_{\pm 0.63}$ | $35.48_{\pm 0.71}$ |
| 16.8% | $34.26_{\pm 0.61}$ | $34.22_{\pm 0.63}$ |
| 13.4% | $33.08_{\pm 0.53}$ | $33.07_{\pm 0.59}$ |
| 10.7% | $31.90_{\pm 0.45}$ | $31.91_{\pm 0.62}$ |
| 8.6% | $30.78_{\pm 0.50}$ | $30.80_{\pm 0.66}$ |

Figure 7: Comparison of the average training PSNR of meta-learned SIRENs after 100 steps of per-signal training. Comparing with Fig. 2, we observe that the PSNR on training and test data are roughly similar.

# E    Experimental result with additional meta learning method

We additionally test Meta-SparseINR with the weights learned by Reptile [30] instead of MAML [5], and arrive at the same conclusion that it outperforms the dense/randomly-pruned baselines, with a slightly smaller PSNR gap ($\approx 2$). This suggests that the benefit of the proposed pipeline is not limited to the case where we use MAML. Fig. 8 is the plot for the Reptile on the CelebA dataset.

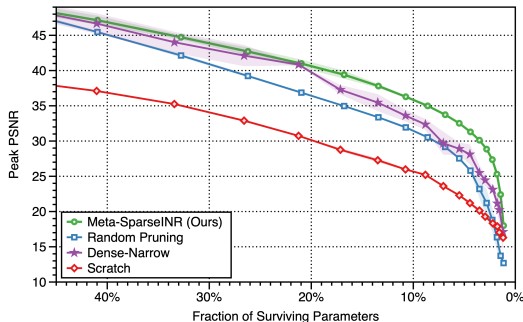

Figure 8: Comparison of the average PSNR of meta-learned SIRENs with Reptile on CelebA after $500$ steps of per-signal training. Shaded region: Mean $\pm$ one standard deviation (three seeds).

# F    Comparison with data compression schemes

Our scheme can also be interpreted as a data compression scheme (while not being optimized for the purpose). To this end, we have compared the compression performance (measured in bits per pixel[6]) of Meta-SparseINR with JPEG in Fig. 9. Perhaps not surprisingly, the compression performance of Meta-SparseINR is worse than JPEG. However, we suspect that the performance of Meta-SparseINR could be further improved by performing a search over the width and depth of the original network, as suggested by Dupont et al. [4].

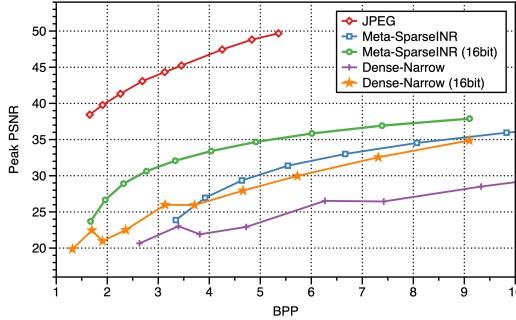

Figure 9: Comparison of the PSNR of JPEG compression and meta-learned SIRENs after $100$ steps of per-signal training. 16bit denotes the converted 16bit percision model which is originally trained as 32bit precision.

---

[6]bits-per-pixel$= \frac{\#\text{parameters} \times \text{bits-per-parameter}}{\#\text{pixels}}$

# G  Cross dataset generalization

We also perform additional experiments under the cross-dataset generalization scenario where the train and test dataset differs. To this end, we have tested two setups: adapting the compressed model trained on CelebA to the Imagenette dataset, and vice versa. As in Fig. 10, Meta-SparseINR continued to outperform the baselines (as much as $\approx 5$ in PSNR), showing a similar gap as in the in-domain experiments.

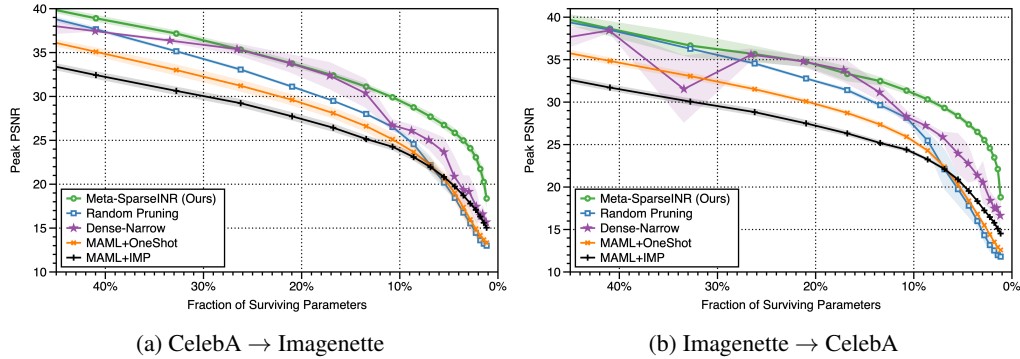

(a) CelebA $\rightarrow$ Imagenette          (b) Imagenette $\rightarrow$ CelebA

Figure 10: Comparison of the average PSNR of meta-learned SIRENs after 100 steps of per-signal training on cross-dataset scenarios. Shaded region: Mean $\pm$ one standard deviation (three seeds).