# OpenReview forum: "Meta-Learning Sparse Implicit Neural Representations"
_NeurIPS.cc/2021/Conference — NeurIPS 2021 Poster_

### Official Review · Reviewer_kwH6 · 2021-07-13

**Rating:** 7
**Confidence:** 4

**Summary:**

Authors propose a method for learning sparse implicit functional representations for signal data. `Their method combines ideas from MAML to learn a good initialization from which a sparse neural network representing the signal can be efficiently estimated. Experiments on a few datasets suggest that the proposed method achieves better PSNR than more naive approaches for the same levels of sparsity. Overall, the paper is in a good shape but still could benefit from a few additions.

**Limitations And Societal Impact:**

Discussed adequately.

**Main Review:**

Novelty:
To my knowledge, both the application and the proposed method are novel.

Clarity:
Overall the paper is well-written and easy to follow.

A minor comment is that I would prefer more rigour in presenting eq. 4 and connecting it to algorithm 1, particularly, in how the iterative procedure is actually performing the inner-loop optimization explicitly writtein eq. 4. Does the algorithm at least provably improves on the objective? It is not completely obvious to me and I would like to know exactly what guarantees Algorithm 1 has.

Significance:

To me it's fairly interesting to be able to sparsely represent natural signals both from abstract point of view and also because it's opening certain applications such as more efficient image or audio compression algorithms.

I'm this very curious if there is any chance of comparing their method to standard image compression algorithms such as JPEG in the paper which should not be very difficult.

Quality:

Overall I find the proposal method technically sound and experimental evaluation adequate (also see my comment in the clarity section).

One of the baselines which I think is missing from the experimental evaluation is some kind of a functional/implicit-style auto-encoder with the varying dimensionality of the hidden variable. I understand that it won't be fully comparable to the model because even if the code itself is short, the decoder network still needs to be transmitted somehow, but arguably one can do this beforehand, so I still think it is a valid experiment (also see question 1).

I also don't fully understand if a more standard lottery-ticket method can applied for the task of simply learning a sparse representation of a single signal. Albeit, again, not being fully comparable to the proposed method, it would probably achieve even better PSNRs and thus provide an important, perhaps gold-standard data-point.

Other questions:
1) Have authors tried an alternative approach where the neural network representing the function is not sparse itself, but only a few parameters of it are allowed to be modified and supposed to contain information about a particular signal?
An example of this approach be found in [1] (although there the setting was quite different), where certain weights in some of the fully-connected layers are modifyable.
2) Why did authors use Imagenette and not the standard ImageNet? I would be more confident (perhaps, falsely) in the results if they were performed on a more diverse dataset. I don't think this would require much work compared to Imagenette experiments and I'm very curiuous to see the results.

References:
[1] Bartunov, S., Rae, J. W., Osindero, S., & Lillicrap, T. P. (2019). Meta-learning deep energy-based memory models. arXiv preprint arXiv:1910.02720.

**Time Spent Reviewing:**

5

---

> ### Author Response · Authors · 2021-08-10
> **Response to reviewer kwH6**
>
> Dear reviewer kwH6,
>
> Thank you for your positive and constructive feedback. In what follows, we respond to your comments.
>
> ---
>
> __Comparison with Data Compression schemes.__
>
> First, we would like to remind that the primary purpose of our paper is to provide a method to “efficiently train compressed INRs,” instead of providing a method to compress the data maximally; we focus on providing model parameters that can be directly used for inference without any decoding step, yet having a smaller size. In fact, if we focus solely on the storage-efficiency, we can perform additional Huffman coding on the weight parameters, sacrificing usability for storage efficiency.
>
> On the other hand, as the reviewer pointed out, our scheme can also be interpreted as a data compression scheme (while not being optimized for the purpose). To this end, we have compared the compression performance of Meta-SparseINR with JPEG. Following link contains the comparison plot: __[LINK](https://drive.google.com/file/d/1gvYlKLiiLXAnKAwmmxxCffLHGoJQE_1T/view?usp=sharing)__
>
> Perhaps not surprisingly, the compression performance of Meta-SparseINR is worse than JPEG. However, we suspect that the performance of Meta-SparseINR could be further improved by performing a search over the width and depth of the original network, as suggested by Dupont et al. (2021).
>
> _Dupont et al., “COIN: Compression with implicit neural representations,” arXiv 2103.03123v2 (2021)_
>
> ---
>
> __On Eq.(4) and Algorithm 1.__
>
> Thank you for this suggestion. We will add more details in Section 3 to make the connection between Eq.(4) and Algorithm 1 more concrete, including providing references to the existing guarantees on gradient-based meta-learning algorithms that we mainly use, e.g., Khodak et al. (2019).
>
> _Khodak et al., “Provable guarantees for gradient-based meta-learning,” ICML 2019._
>
> ---
>
> __Modification-Sparsity instead of Parameter-Sparsity (Bartunov et al. 2020).__
>
> We are happy that you pointed out this alternative. As we briefly mentioned in Section 6 of our manuscript, we also think that storing sparse differences for each sample (possibly with different sparsity or sparsity patterns for each sample) is a promising future direction to pursue, especially when our central concern is the storage efficiency instead of compute or inference efficiency. Actually, we have been working on this idea as a follow-up study, but unfortunately have not seen much success yet. None of the tried methods---including the ones using the SNIP (Lee et al., 2019) or movement pruning (Sanh et al., 2020) saliency scores--- consistently outperformed the parameter-sparsity (i.e., Meta-SparseINR). This rather suggests that our method, Meta-SparseINR, can be a strong, not-easy-to-beat baseline for the future studies on related topics.
>
> We thank the reviewer for the pointer to Bartunov et al. (2020); we would try exploring this direction as well.
>
> _Lee et al., “SNIP: Single-shot network pruning based on connection sensitivity,” ICLR 2019._
>
> _Sanh et al., “Movement pruning: Adaptive sparsity by fine-tuning,” NeurIPS 2020._
>
> ---
>
> __Imagenette not ImageNet.__
>
> We used Imagenette (which is a subsampled version of ImageNet) following the setup of Tancik et al. (2021), which is the paper where the method for Dense-Narrow baseline is introduced.
>
> We agree with the reviewer’s point that training on a more diverse ImageNet may benefit the readers. Unfortunately, however, the time required for running the ImageNet experiment was too long to be completed before the end of the response period, given the scale of computational resources we have; we need a much larger number of meta-training steps, which should be iterated multiple times for pruning. We would aim to provide the result until the final version.
>
> _Tancik et al., “Learned initializations for optimizing coordinate-based neural representations,” CVPR 2021._
>
> ---
>
> __Standard lottery ticket method.__
>
> As we showed in Section 5.3, the lottery ticket methods failed to outperform training a dense-narrow model from scratch. Due to this reason, we chose not to report the lottery ticket baseline in the main experiment section.
>
> ---
>
> We hope that our response addressed your concerns reasonably. Please let us know if you have any further questions.
>
> Best,
>
> Authors.

---

### Official Review · Reviewer_mx13 · 2021-07-13

**Rating:** 7
**Confidence:** 4

**Summary:**

The paper describes a method for representing sets of signals using coordinate-based neural representations (or INRs) by combining a meta learning approach for learning the initial weights with a network pruning approach for ensuring that this initialization is as sparse and memory-efficient as possible. The paper contributes an algorithm, Meta-SparseINR, which alternates between meta-learning weight initializations for coordinate-based networks and pruning the weights in the meta-learned initialization. The algorithm results in a sparse network initialization which can be then specialized in a few gradient steps to represent any signal in a dataset, and generalize to new signals taken from the same distribution. The paper demonstrates that this additional pruning step built into the meta learning process leads to improved results when compared to just meta-learning dense network initializations with a comparable number of parameters. The method is evaluated on an image-fitting task for multiple datasets, and it is shown that this approach does lead to a better performance versus memory trade-off when compared to just network pruning or meta-learning alone.


**Ethical Concerns:**

In my opinion, there are no ethical issues with this paper.

**Limitations And Societal Impact:**

The authors have adequately addressed the limitations and potential negative societal impacts of their work.

**Main Review:**

I will break down the review into an analysis of strengths and weaknesses of the paper. In my opinion, the strengths of the paper are:
1. The results show that applying the combination of meta learning and network pruning iteratively as in Meta-SparseINR result in noticeable improvement over performing either of these methods alone. This is a valuable contribution, since it justifies the papers’ claim that this method can bridge the gap between meta learning methods which speed up learning INRs, and quantization or pruning methods which decrease the memory required to store a dataset of INRs.
    - The experiments show that Meta-SparseINR outperforms just performing meta learning with a representation with the same number of parameters, especially when there are few parameters (Fig 2). Interestingly, the sparse representation also converges with less optimization steps (Fig 3a).
    - The experiments show that Meta-SparseINR retains the benefits of meta learning and converges significantly faster to a higher quality representation than training from scratch (and presumably pruning or quantization this scratch training, since this likely does not improve the quality of the representations trained) (Fig 3a).
2. The paper is well motivated - one of the largest challenges with coordinate-based networks is generalization to represent datasets of signals. This paper attempts to remedy this by considering both axes of computational requirements to do this - optimization time, and memory consumption. This could potentially enable representing large datasets with INRs, which would allow for easier studying of the characteristics of (and possibly generalization across) these types of signal representations.
3. The paper is written very clearly and concisely. The experiments shown directly justify the claims that are made in the paper, especially those in Fig 2, 3, 4. The methodology is described in detail, and justification for design choices is provided. This is valuable as it makes it much easier to understand what the method is doing, and thus more likely to be adopted into future research work.

The weaknesses of the paper are:
1. The main concern I have is related to the choice of learning a sparse initialization, instead of learning an initialization which can then be made sparse when fit to each dataset signal. I have a few questions and possible concerns:
    - The method only learns a sparse initial INR. Once this initialization is applied to represent a new signal, is the learned mask retained and the final represented signal also sparse? It must be, otherwise memory is only saved when storing the **initialization**, and not each of the dataset signal representations.
    - The most logical baseline to compare to would be one which first meta-learns an initialization (non-sparse), and then uses this initialization to fit each of the dataset signals independently (non-sparse), and then applies some pruning or quantization method to each of these final representations (sparse or quantized). In this case, the (non-sparse) meta-learned initialization would still provide the same speedup bonus on optimization, and the quantization or pruning on the (non-sparse) dataset signal representation could provide a similar memory benefit in storage by making it sparse or quantized. I understand that pruning is an iterative process, which requires fine tuning after making a representation sparse, so this type of baseline would likely still incur additional computational cost when fitting each dataset signal for this fine tuning. But, I still think this makes sense to compare to, in terms of computational requirement and representation quality, alongside just simply applying meta learning to a smaller INR architecture.
    - Without some comparison like this or explanation why this baseline doesn’t make sense, I think that it’s not super well justified why the meta learning and pruning steps need to be iteratively applied to the initialization, instead of sequentially applied to the initialization and then the dataset signals. Since this is the main originality of the contributed algorithm, I think some justification of this would make the paper significantly stronger.
2. I think there is a flaw in the conclusion of section 5.3. Figure 5 shows that pruning using the *Winning Ticket* method performs equivalently to just training a smaller representation, which is supposed to highlight some benefit of sparsity when trained with the meta-initialization. However, it is explained that these results are obtained by training each of these representations for 10k steps, while the comparable plot where the sparse initializations outperform the smaller networks is only benchmarked after training for 100 iterations (Fig 2). I suspect that meta-learning only enables improved performance for a limited number of iterations, after which the magnitude of contribution of the initialization likely doesn’t matter anymore. I.e., if Fig 2 was left to train for as long as Fig 5, it’s possible that we’d see the same trend with the meta-learned initializations as well.

Overall, the paper contributes a valuable algorithm for combining network pruning with meta-learning methods. The contribution of this algorithm is well-motivated for learning to represent classes of signals using INRs, and the paper shows that it does better than existing methods which apply just meta learning or pruning. Because of this, I think the paper is already quite good. I believe that the comparison to independently performing meta-learning **and** network pruning could remove all reasonable doubt that the formulation of Meta-SparseINR for learning the initialization, and then directly optimizing this sparse representation on dataset signals, is the right way to go about learning a dataset of INRs. This, along with some clarification on the conclusions drawn from Fig 5, would make this a much stronger paper.

Some other minor comments which could possibly be clarified to improve the quality of the paper:
- For Fig 5, the supplement mentions that this was obtained from a single image of a Fox? Does this trend hold for multiple images.
- Can the repeated application of sparsity throughout the meta learning process reduce the memory required to train the model? One big issue with MAML and other gradient-based meta learning algorithms is the memory requirement of differentiating through the optimization process. If making this network sparser can reduce this, this could be very impactful.

**Update after Author Response**:
The additional experiments provided by the authors have addressed the concerns brought up in my original review. I retain my score, and think that this is a good paper which should be accepted.


**Time Spent Reviewing:**

3.5

---

> ### Author Response · Authors · 2021-08-10
> **Response to reviewer mx13**
>
> Dear reviewer mx13,
>
> Thank you for your insightful and detailed review. Below, we respond to your comments.
>
> ---
>
> __Is the final model also sparse?__
>
> Yes, the final model---trained on each signal from Meta-SparseINR---is also sparse. In other words, the learned mask is retained. We will clarify this in the abstract and introduction of the final version.
>
> ---
>
> __Comparison with MAML-then-Prune baseline.__
>
> We agree with the reviewer’s point that simply pruning the MAML-trained method is a logical baseline to compare with, to evaluate the benefit of our combined meta-learning and pruning pipeline. To this end, we tested the following two additional baselines that are designed to have the same/comparable per-signal computational budget (100 training epochs) as other baselines and Meta-SparseINR:
> - _MAML+OneShot:_ We train the meta-learned model on each signal for 50 epochs, perform magnitude-based pruning, and run additional 50 epochs.
> - _MAML+IMP:_ We prune the model $t$ times, removing 20% of the surviving weights each time, and running $\frac{100}{t+1}$ epochs before/between/after each pruning phase.
>
> From the experiment, we observe that Meta-SparseINR outperforms both baselines by the PSNR difference $\approx 4$. The new baselines perform similarly to Random Pruning, and worse than Dense-Narrow. Following link contains experimental results on CelebA and Imagenette dataset with the baselines (3 seeds): __[LINK](https://drive.google.com/file/d/19LO26_ZwrOYstmJMKbABloyjbXZe3XaY/view?usp=sharing)__
>
> These baselines will be added to the final version.
>
> ---
>
> __Lottery ticket experiment: Conclusion.__
>
> Thank you for pointing this out. Per-signal training budget for the lottery ticket experiment was indeed set much longer than that of the main experiment. On one hand, this is because our aim was to measure whether the benefit of lottery tickets holds after full training, similarly to how lottery tickets are usually evaluated for most non-INR setups. On the other hand, as the reviewer pointed out, to rigorously compare the effect of sparsity with/without meta-learned initialization, we need to know (1) how meta-learned sparse INRs perform after long training, and (2) whether lottery tickets are advantageous during the early stage.
>
> To this end, we performed additional experiments, and observed that (1) meta-learned sparse INRs indeed have benefits over Dense-Narrow after an extended amount of training ($\sim$50,000 epochs), and (2) lottery tickets do have some benefits during the early stage of training, but their benefits over dense-narrow models quickly vanish as the PSNR starts to reach over 20 ($\sim$1,000 epochs).
>
> Following table summarizes (1), for two different levels of sparsity and two datasets (3 seeds).
> $
> \begin{array}{lcccc}
> \text{Dataset} & \text{CelebA} & & \text{Imagenette} & \newline
> \text{Surv. Weights} & \text{10\\%} & \text{5\\%} & \text{10\\%} & \text{5\\%} \newline
> \hline
> \text{Meta-SparseINR} & 40.62{\scriptsize \pm 0.93} & 35.87{\scriptsize \pm 0.57} & 37.31{\scriptsize \pm 0.38} & 32.52{\scriptsize \pm 0.27} \newline
> \text{Dense-Narrow} & 36.65{\scriptsize \pm 2.50} & 31.09{\scriptsize \pm 2.67} & 33.27{\scriptsize \pm 1.75} & 27.64{\scriptsize \pm 3.18}
> \end{array}
> $
>
> Following link contains the performance plots for lottery tickets after 100 and 1000 steps of training on SIREN on the Fox image, which corresponds to the experiment (2). (3 seeds): __[LINK](https://drive.google.com/file/d/1dpeZjXyhOKd3f89U-4wD7GeUvSbf-SuS/view?usp=sharing)__
>
> In the final version, we will enhance and revise section 5.3 (and introduction) to reflect these findings.
>
> ---
>
> __Lottery ticket experiment: Other than “Fox”.__
>
> Following the reviewer’s suggestion, we performed additional experiments on the Kodak dataset, and we observed that the same trend holds. Following link contains the plot for SIREN on the Kodak dataset (1 seed): __[LINK](https://drive.google.com/file/d/156azOKUziBQe8Adp4hgSU5XAOFmWAwIc/view?usp=sharing)__
>
> The three-seed version of these experiments will be added to section 5.3 in the final version.
>
> ---
>
> __Does sparsity help resolve memory issues with MAML.__
>
> Ideally (or theoretically), the repeated application of sparsity can help reduce the memory footprint of meta-learning INRs. To fully enjoy the benefits of sparsity during meta-learning, however, we would need to use additional kernel-level tricks, such as n:m sparsity (Mishra et al., (2021)). Indeed, resolving the computational bottlenecks of meta-learning via sparsity seems to be a promising yet underexplored direction.
>
> _Mishra et al., “Accelerating sparse deep neural networks,” arXiv 2104.08378v1 (2021)._
>
> ---
>
> We hope that our response addressed your concerns reasonably. Please let us know if you have any additional questions.
>
> Best,
>
> Authors.

---

### Official Review · Reviewer_oRre · 2021-07-16

**Rating:** 5
**Confidence:** 4

**Summary:**

This paper presents a method of alternately meta-learning the Implicit Neural Representations (INR) for a set of signals using the MAML method, and pruning the INR using the global magnitude-based pruning method for sparsifying the INR. The proposed method is tested on the 2D image regression task in three datasets (CeleA, Imagenette and 2D SDF). In comparisons, three baselines are used (MAML+w/ random pruning,  MAML+w/ dense-narrow, and w/o MAML + dense-narrow). The proposed method obtains better PSNR across different number of optimization steps, thus showing more efficient learning in terms of computation and memory footprint.

**Limitations And Societal Impact:**

The limitations of the proposed method are considered in the manuscript, but not sufficient as stated above. Potential ethical side-effects are addressed.

**Main Review:**

Overall, the proposed method seems to apply existing techniques, a meta-learning method using MAML and a pruning strategy using global magnitude-based pruning, to sparsify the INRs.  The technical novelty thus seems to be limited.  The baselines also are limited. It will be better If different meta-learning methods and pruning strategies are tested.  So, I think this paper is marginally below the bar of NeurIPS.

More specifically,

Pros:
+ The paper is well motivated and easy to follow.
+ The observations of applying MAML+Magnitude-based Pruning in sparsifying INRs are informative.
+ The experimental results are promising with good analyses.

Cons:
- Only one type of meta-learning methods, MAML and one type of pruning strategies, magnitude-based pruning are studied. More are entailed to gain more insights.
- Vanilla INRs are used (SIREN and FFN). It will be better if some new designs are studied towards more effective and more efficient meta-learning and pruning.
- The proposed method does not test the cross-dataset meta-learning and pruning, which seems to be more interesting for scaling up INRs in a computation and memory efficient way

**Time Spent Reviewing:**

about 3

---

> ### Author Response · Authors · 2021-08-10
> **Response to Reviewer oRre**
>
> Dear reviewer oRre,
>
> Thank you for taking your time to provide constructive feedback, and acknowledging the clarity of our motivation and effectiveness of our method. We address your concerns below.
>
> ---
>
> __Applying existing techniques?__
>
> We re-emphasize that our central contribution lies in the design of the compression pipeline itself. Our approach (illustrated in Figure 1) finds a solution efficiently in terms of both time and memory, which stands in stark contrast to conceivable, yet naïve alternatives with existing techniques in terms of the efficiency and the final performance, as noted by other reviews; reviewer mx13 noted that our method “can bridge the gap between meta-learning methods (...) and quantization or pruning methods” by “considering both axes of computational requirements,” and reviewer kwH6 found that “both application and the proposed methods are novel.” Specifically, the proposed pipeline prunes the meta-learned model before fitting each signal, eliminating the need to perform time-consuming iterative pruning on every individual signal.
>
> In fact, we show that our method significantly outperforms the naïve method of independently performing MAML and magnitude-based pruning. To this end, we evaluated the following two additional baselines that are designed to have the same/comparable per-signal computational budget (100 training epochs) as other baselines and Meta-SparseINR:
> - _MAML+OneShot:_ We train the meta-learned model on each signal for 50 epochs, perform magnitude-based pruning at a single shot, and run additional 50 epochs of training.
> - _MAML+IMP:_ We prune the model $t$ times, removing 20% of the surviving weights each time, and running $\frac{100}{t+1}$ epochs before/between/after each pruning phase.
>
> From the experiment, we observe that Meta-SparseINR outperforms both baselines by the PSNR difference $\approx 4$. The new baselines perform similarly to Random Pruning, and worse than Dense-Narrow. Following link contains experimental results on CelebA and Imagenette dataset with the baselines (3 seeds): __[LINK](https://drive.google.com/file/d/19LO26_ZwrOYstmJMKbABloyjbXZe3XaY/view?usp=sharing)__
>
> These results will be added to the final version.
>
> ---
>
> __Meta-learning algorithms other than MAML.__
>
> We additionally tested Meta-SparseINR with the weights learned by Reptile (Nichol et al., 2018) instead of MAML, and arrived at the same conclusion that it outperforms the dense/randomly-pruned baselines, with a slightly smaller PSNR gap ($\approx 2$).  This suggests that the benefit of the proposed pipeline is not limited to the case where we use MAML. Here is the plot for the Reptile on the CelebA dataset (3 seeds): __[LINK](https://drive.google.com/file/d/1hw7VCwup8N3_ioR0iNbIP8VrYyddno9J/view?usp=sharing)__
>
> The results will be added to the final version. Thank you very much for the suggestion.
>
> _Nichol et al., “On first-order meta-learning algorithms,” arXiv 1803:02999 (2018)._
>
> ---
>
> __Pruning methods other than MP.__
>
> We chose MP for two non-trivial reasons: (1) MP is known to achieve state-of-the-art performance (see Gale et al. (2019)), and (2) MP is free from computing (the first-/second-order) loss gradients. Especially for the latter, we note that pruning methods that require gradients in their saliency criteria cannot be evaluated unless one re-designs pruning algorithms based on meta-learning losses; computing higher-order derivatives in addition to what is required for meta-level differentiation amounts too much compute burden. Thus, we do not find them appropriate to be an immediate “baseline” to compare with. We will make these points clear in the final version.
>
> _Gale et al., “The state of sparsity in deep neural networks,” arXiv 1902.09574 (2019)_
>
> ---
>
> __Using vanilla INRs, instead of studying new designs.__
>
> We do not necessarily view the fact that our paper considers pruning vanilla INRs, instead of proposing a new model design, as our weakness. Instead, we view two lines of study as complementary to each other. For example, in the image classification literature, studies on model pruning (e.g., Han et al. (2015)) and efficient model design (e.g., Tan & Le (2019)) have developed as independent lines of work, and many works aim to enjoy the benefit of both by compressing the efficient models even further, e.g., Azarian et al. (2020).
>
> Our work initiates a study of model compression under the INR context, which we hope to be jointly used with efficient INR model architectures for a better efficiency.
>
> _Han et al., “Learning both weights and connections for efficient neural networks,” NeurIPS 2015._
>
> _Tan & Le, “EfficientNet: Rethinking model scaling for convolutional neural networks,” ICML 2019._
>
> _Azarian et al., “Learned threshold pruning,”arXiv 2003.00075 (2020)._
>
> ---
>
> __Cross-dataset.__
>
> Following the reviewer’s suggestion, we have performed additional experiments under the cross-dataset scenario. In particular, we have tested two setups: adapting the compressed model trained on CelebA to the Imagenette dataset, and vice versa. In the experiments, Meta-SparseINR continued to outperform the baselines (as much as $\approx 5$ in PSNR), showing a similar gap as in the in-domain experiments. Following link contains the plot (3 seeds): __[LINK](https://drive.google.com/file/d/1JmxXtqtq_idx_FH9u20o_oL2NjDNckoN/view?usp=sharing)__
>
> These results will be added to the final version, which we do believe strengthens our paper.
>
> ---
>
> We hope that our answers resolved your concerns on our contributions and baselines. Please let us know if you have any unclear parts; we would be more than happy to discuss further.
>
> Best,
>
> Authors.

---

> > ### Comment · Reviewer_oRre · 2021-08-18
> > **My concerns have been addressed**
> >
> > Dear authors,
> >
> > Thank you for your rebuttal. Most of my concerns have been addressed. Thanks

---

> > > ### Author Response · Authors · 2021-08-19
> > > **Thank you for the response.**
> > >
> > > Thank you for letting us know; we are happy to hear that our rebuttal addressed your concerns well.
> > >
> > > If you have any remaining suggestions or concerns, please let us know!
> > >
> > > Best, Authors.

---

### Author Response · Authors · 2021-08-18
**A gentle reminder**

Dear Reviewers,

Thank you again for your time and efforts in reviewing our paper.

We kindly remind that we are more than one week into the discussion period. We believe that we sincerely and successfully address your concerns/questions/misunderstandings/suggestions, with the results of the supporting experiments.

If you have any further concerns or questions, please do not hesitate to let us know.

Thank you very much!

Authors

---

### Decision · Program_Chairs · 2021-09-27

**Decision:**

Accept (Poster)

**Comment:**

The paper proposes to learn functional representations of signals such as images using meta learning and network pruning, so that an initial sparse network can be perturbed to fit each training or testing image. The proposed method is novel, simple and clean, and it can be useful in applications based on functional representations. The experiments are solid and thorough, especially with the additional experiments performed during the rebuttal period.

After rebuttal, this paper received three final ratings of 7, 7, 7 (the reviewer who initially assigned a rating of 5 upgraded the rating to 7, but was unable to make the change formally due to technical error). The rebuttals have addressed all the concerns of the reviewers.

This paper makes a valuable contribution and can be accepted.